# Risk Factors Associated with 30-Day Mortality in Older Patients with Influenza

**DOI:** 10.3390/jcm10163521

**Published:** 2021-08-11

**Authors:** Charles Guesneau, Anne Sophie Boureau, Céline Bourigault, Gilles Berrut, Didier Lepelletier, Laure de Decker, Guillaume Chapelet

**Affiliations:** 1Clinical Gerontology Department, Nantes University Hospital, 1 Place Alexis-Ricordeau, F-44000 Nantes, France; charles.guesneau@chu-nantes.fr (C.G.); annesophie.boureau@chu-nantes.fr (A.S.B.); gilles.berrut@chu-nantes.fr (G.B.); laure.dedecker@chu-nantes.fr (L.d.D.); 2Université de Nantes, EE MiHAR (Microbiotes, Hôtes, Antibiotiques et Résistance Bacterienne), Institut de Recherche en Santé (IRS2), 22 Boulevard Bénoni-Goullin, F-44200 Nantes, France; celine.bourigault@chu-nantes.fr (C.B.); didier.lepelletier@chu-nantes.fr (D.L.); 3Bacteriology and Infection Control Department, Nantes University Hospital, 1 Place Alexis-Ricordeau, F-44000 Nantes, France

**Keywords:** influenza, elderly, risk factors, mortality

## Abstract

Background: Influenza is a common viral condition, but factors related to short-term mortality have not been fully studied in older adults. Our objective was to determine whether there is an association between geriatric factors and 30-day mortality. Methods: This was a retrospective cohort design. All patients aged 75 years and over, with a diagnosis of influenza confirmed by a positive RT-PCR, were included. The primary endpoint was death within the 30 days after diagnosis. Results: 114 patients were included; 14 (12.3%) patients died within 30 days. In multivariate analysis these patients were older (OR: 1.37 95% CI (1.05, 1.79), *p* = 0.021), and had a lower ADL score (OR: 0.36 95% CI (0, 17; 0.75), *p* = 0.006), and a higher SOFA score (OR: 2.30 95% CI (1.07, 4.94), *p* = 0.03). Oseltamivir treatment, initiated within the first 48 h, was independently associated with survival (OR: 0.04 95% CI (0.002, 0.78), *p* = 0.034). Conclusions: Identification of mortality risk factors makes it possible to consider specific secondary prevention measures such as the rapid introduction of antiviral treatment. Combined with primary prevention, these measures could help to limit the mortality associated with influenza in older patients.

## 1. Introduction

It has been estimated that the global prevalence of influenza is around 1 billion cases annually. Of these, 3–5 million cases are severe, causing between 290,000 and 650,000 deaths (i.e., 4.0–8.8 per 100,000 individuals) [1], especially among people aged 75 years or older (51.3–99.4 per 100,000 individuals) [1]. In France, the incidence of deaths exceeds 9000 per season, corresponding to 11% of all-cause deaths during the epidemic season in the entire population [2].

Mortality is higher in elderly patients, and influenza-related deaths increase exponentially after the age of 65 [3]. However, few studies have highlighted the impact of specific geriatric criteria on short-term mortality. Several studies have identified variables associated with influenza-related mortality [4,5,6,7,8,9,10,11,12,13,14], but very few have focused specifically on elderly patients [15,16,17]. In these studies, the following factors were associated with short-term mortality: age [6,12]; presence of underlying diseases [13] including diabetes [5,8] immunosuppression [4,7,8,17], chronic respiratory disease [6,7], chronic cardiac disease [6,14], renal disease [6], and cancer [7,12]; number of drugs being administered [6,16]; malnutrition [16]; recent infectious disease [16]; severity of illness [4,5,10,12,14,17] including bacterial co-infection [9] and nosocomial infection [9]; delays in adequate treatment and admission to the intensive care unit [7,8,11]; delays in the initiation of antiviral treatment (within the first 48 h) [8,9,13,17]; and number of consultations in the past year [15].

Our hypothesis was that geriatric factors are associated with short-term mortality in elderly patients with seasonal influenza. We analyzed data during the epidemic period between January and March 2015, when an increased prevalence of influenza was described by the European Influenza Surveillance Scheme (EISS) [18]. Our objective was to determine whether there was an association between geriatric variables and 30-day mortality in patients with seasonal influenza.

## 2. Materials and Methods

### 2.1. Study Population

This retrospective cohort study was conducted in 2017 in a tertiary, 2600-bed, university-affiliated center at the Clinical Gerontology Department, Nantes, France. The Clinical Gerontology Department is composed of 2 acute-care services, 2 post-acute care services, and 2 nursing home residences.

From January to April 2015, all patients who were 75 years of age or older were retrospectively included in this study if they met all the following criteria: (i) influenza-like symptoms (fever and/or respiratory signs and/or myalgias and/or headache); (ii) positive RT-PCR (detection of influenza A or B); and (iii) no admission to ICU. For each patient included, only the first episode of positive RT-PCR was considered. Patients who did not meet the detailed definition of the inclusion criteria were not included.

### 2.2. Data Collection

Our primary endpoint was presence/absence of death within the 30 days after the diagnosis of influenza to investigate factors associated with short-term mortality. The analyses of medical records and 2 death notification registers (www.avis-de-deces.net; www.dansnoscoeurs.fr) were considered to assess the primary outcome. General practitioners were called if the outcome was not available in the medical record and the death notification registers.

Considering the results of the previous studies focused on short-term mortality in general population [4,5,6,7,8,9,10,11,12,13,14] and in elderly patients [15,16,17], the following information was collected concerning patient’s characteristics: age; sex; presence of an underlying disease including diabetes, heart disease, chronic lung disease, chronic obstructive pulmonary disease (COPD), renal disease, cancer, and hypogammaglobulinemia; the Charlson’s Comorbidity Index score [19]; the Katz Index of Independence in Activities of Daily Living (ADL) [20]; the number of drugs being administered including statins/immunosuppressive/corticoids; and influenza vaccination status. Moreover, concerning influenza infection the following variables were collected: detection of influenza A or B viruses by RT-PCR; date of sampling and delay (in days) between diagnosis and the onset of symptoms; hospitalization or admission to an intensive care unit; nosocomial influenza infection (infection acquired at least after 48 h of hospitalization in the acute care services or in the post-acute care service); severity of infection i.e., SOFA score at the diagnosis [21]; prescription of antiviral treatment, including the delay between the onset of symptoms and antiviral prescription; and antibiotic co-prescription.

Considering previous studies that had highlighted factors associated with short-term mortality, “chronic respiratory disease” corresponds to patients with history of chronic lung disease and/or chronic obstructive pulmonary disease (COPD), and “immunosuppression” corresponds to patients with active cancer and/or hemopathy, or history of cancer and/or hemopathy in the last 5 years, and/or hypogammaglobulinemia, and/or treatment with anti-inflammatory and/or immunosuppressive drugs, including corticosteroids.

### 2.3. Statistical Analysis

The participant’s baseline characteristics were summarized using mean and standard deviations or frequencies and percentages, as appropriate. The analyzed variable of interest was short-term mortality, i.e., mortality in the 30 days following the diagnosis of infection. Between-group comparisons were performing using an independent simple t-test or chi-squared test (X2), as appropriate. The Shapiro–Wilk test was used to assess the normality of continuous variables. Non-normally distributed continuous variables were compared using the Mann–Whitney test. Normally distributed continuous variables were compared using Student´s t-test. Univariable and multivariable logistic regressions were performed to examine the association between short-term mortality and other variables. Variables with a significant association in univariate analysis and/or with a *p*-value < 0.2 were entered into a logistic regression model for multivariate analysis. Relative risks were expressed as odds ratios (OR) and 95% confidence intervals. All reported *p*-values < 0.05 were considered as statistically significant. Analysis was performed using SPSS software version 15.0 (SPSS, Inc., Chicago, IL, USA).

## 3. Results

### 3.1. Patient Characteristics

From January to March 2015, 122 nasopharyngeal swabs tested positive for the detection of the influenza virus by RT-PCR at the Clinical Gerontology Department (Figure 1): 115 (94.3%) were positive for influenza A, and 7 (5.7%) for influenza B. One patient had two positive RT-PCR results. For this patient, we considered only the first episode (influenza B detection by RT-PCR), but not the second episode (influenza A detection by RT-PCR). Four patients had positive RT-PCR results but without influenza-like symptoms. Three patients were aged under 75 years old. Finally, 114 patients were included. Positive RT-PCR distributions over time are detailed in Appendix A. The repartition of positive results according to the different units of the Clinical Gerontology Department are detailed in Appendix A. Patient characteristics (*n* = 114) are detailed in Table 1; 82 (71.9%) were female, median age ( ± SD) 87.9 ± 5.5. The mean Charlson Comorbidity Index score (± SD) was 2.83 (± 1.7), and the mean ADL score (± SD) was 3.08 (± 2.1).

### 3.2. Factors Associated with Short-Term Mortality

The 30-day mortality incidence was 12.3% (14/114). The characteristics of survivors and non-survivors are shown in Table 1. The comparison between groups revealed significant differences concerning the following factors: age (*p* = 0.006); mean ADL score (*p* = 0.020); mean number of drugs (*p* = 0.026); and mean SOFA score (*p* = 0.001). No differences between groups were observed in terms of morbidity, including the Charlson Comorbidity Index and other influenza infection characteristics.

In univariate analysis (Table 2), the three following factors were significantly associated with short-term mortality: age (OR = 1.18, 95% CI = (1.04–1.35), *p* = 0.013); ADL score (OR = 0.69, 95% CI = (0.05–0.95), *p* = 0.027); and SOFA score (OR = 1.83, 95% CI = (1.27–2.64), *p* = 0.001).

In the multivariate analysis (Table 2), the four following factors were significantly associated with short-term mortality: age (OR = 1.37, 95% CI = (1.05–1.79), *p* = 0.021); ADL score (OR = 0.36, 95% CI = (0.17–0.75), *p* = 0.006); antiviral treatment administration in the first 48 h (OR = 0.04, 95% CI = (0.002–0.78), *p* = 0.034); and SOFA score (OR = 2.30, 95% CI = (1.07–4.94), *p* = 0.034).

## 4. Discussion

Influenza is a common viral condition that can be serious and even fatal in some at-risk populations, including that of older adults. Indeed, mortality is higher in older adults, and influenza-related death increases exponentially after the age of 65 [3]. Few studies have highlighted the impact of specific geriatric factors on short-term mortality. In this study an elevated ADL score and antiviral treatment administration within the first 48 h of symptoms were associated with reduced 30-day mortality, while a high SOFA score and age were associated with increased 30-day mortality.

In the multivariable analysis, age was significantly associated with increased short-term mortality during seasonal influenza epidemics in elderly patients. This result is consistent with previous clinical and epidemiologic studies [1,2,6,12,13] and was confirmed by Mertz et al. in a recent meta-analysis [22]. However, we assume that it could be difficult to distinguish physiological age from chronological age [23], and this result could be biased by confounding factors such as the immunological markers of immunosenescence [24].

In this study, a higher ADL score [20], i.e., a higher ability to performed activities daily living, is associated with a reduced short-term mortality. This result is consistent with the study of Gozalo et al. [16], which highlighted the relationship between influenza and a decline in activities of daily living in nursing home (NH) residents.

In this study, a higher SOFA score was associated with short-term mortality. This result is not surprising because this association was described in a previous study that included ICU patients [12]. Our study confirms this association in an elderly population. Previous studies also found an association between short-term influenza mortality and other severity scores such as the SAPS II score [14] and APACHE score [17]. In this study, we used the SOFA score because its use is consensual in assessing the severity of sepsis [21].

In our study, antiviral (oseltamivir) prescription within the first 48 h was associated with reduced short-term mortality. It was the only modifiable variable associated with mortality in our studies, even if this association could be due to the above and other potential biases and confounders. Although the impact on mortality of antiviral prescription is controversial [25], this result is consistent with previous studies [8,9,13,17]. Antiviral treatment can be prescribed in the elderly population, as was proposed by the Centers for Disease Control and Prevention (CDC) Advisory Committee on Immunization Practices (ACIP) in the two following recommendations [26]: (i) early antiviral treatment prescription is proposed for patients with suspected or confirmed severe influenza (e.g., those who have severe, complicated, or progressive illness or who require hospitalization), (ii) early antiviral treatment prescription is proposed for patients with of suspected or confirmed influenza at higher risk for influenza complications.

In this study, we could not analyze the association between history of influenza vaccination and short-term mortality because of missing data (*n*= 70 (61.4%)). Although influenza vaccination had a positive but modest impact in preventing influenza, its impact on the reduction of mortality is controversial, especially in older patients. The 2010 Cochrane meta-analysis did not conclude that influenza vaccination is effective in people aged 65 and over [27]. However, the critical analysis of this meta-analysis highlighted a vaccine efficacy of 30% for the prevention of lethal and non-lethal influenza complications, 40% for the prevention of clinical influenza, 50% for virologically confirmed influenza, and 60% for the prevention of biological influenza infection [28]. Another study from Fireman et al. [29] highlighted a vaccine effectiveness against all-cause mortality of 4.6% (0.7–8.3), ranging from 5.3% in people aged 65 to 79 to 3.9% in the oldest age group. This effectiveness was estimated at 40% for the prevention of influenza-related deaths. Another meta-analysis conducted in 2018 [27] showed a lower risk of contracting influenza (6–2.4%) and influenza syndrome (6–3.5%) in vaccinated elderly patients. However, the hospitalization and mortality data were insufficient. Finally, influenza vaccination has a significant effectiveness in preventing influenza-related deaths and reducing the risk of hospitalization, but a moderate effectiveness in preventing deaths from all causes. Society should invest in research on a new generation of influenza vaccines for the elderly [27].

We did not observe an association between short-term mortality and the following factors that were previously associated with short-term mortality in patients with influenza: presence of underlying diseases [13] including diabetes [5,8], immunosuppression [4,7,8,17], chronic respiratory disease [6,7], chronic cardiac disease [6,14], renal disease [6], and cancer [7,12]; number of drugs being received [6,16]; malnutrition [16]; recent infectious disease [16] including bacterial co-infection [9]; nosocomial infection [9]; and number of consultations in the past year [15]. Three main methodological biases could explain these results. First, our study may have been underpowered to detect associations. Specifically, nutritional status and oral hygiene could be important factor associated with mortality, and exploration of this aspect was limited due to missing data associated with the retrospective design. Second, in previous studies all confounding factors were not included in the multivariate analysis. Third, our population could be different with regard to the presence and the number of underlying diseases. We hypothesized that this heterogeneity of underlying disease could have biased our results. However, we did not observe an association between mortality and the Charlson comorbidity index score. Finally, we hypothesized that a global score of comorbidities may have less impact than specific comorbidities in short-term mortality, and/or that our study may have been underpowered to detect this.

This study has several limitations. First, this study had an observational design, and we reported the results of a retrospective analysis with a small number of cases, controls, and events (14 deaths). Therefore, this study may have been underpowered for the definitive validation of these results. Hidden bias may exist, with the underestimation of the true relationship between some factors (such as environmental or social factors) and short-term mortality. Second, this was a monocentric study, and the results may not be fully applicable to other settings. However, we assume that our population is representative, and we observed 30-day all-cause mortality. The mortality incidence of 12.3% is consistent with that observed in Europe during this period [18]. Third, all geriatric conditions (such as frailty) could not be included because of the retrospective study design. These factors should be analyzed in further studies. Fourth, risk factors found in one season may reflect an epidemic and not the usual year. Prospective studies should be performed considering multiple years. Fifth, confounding bias by indication could alter the association between oseltamivir treatment and mortality, because the allocation of treatment was not randomized. Sixth, our studies found only one modifiable variable associated with mortality (oseltamivir treatment). However, there is still benefit in quantifying non-modifiable risks for the benefit of prognosis and monitoring.

## 5. Conclusions

This study among elderly patients with influenza highlights that antiviral treatment administration within the first 48 h of symptoms may reduce 30-day mortality, while a high SOFA score and age may increase it. Further studies are necessary to confirm our results and analyze the impact of geriatric conditions, malnutrition, functional or frailty status, and environmental or social factors on short-term mortality in patients with influenzae. These results could help to develop strategies in order to optimize preventive and curative management of influenza in elderly patient.

## Figures and Tables

**Figure 1 jcm-10-03521-f001:**
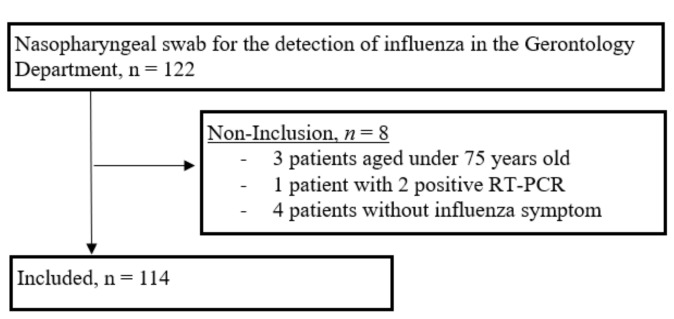
Screening, exclusion, and enrollment process of participants.

**Table 1 jcm-10-03521-t001:** Demographic and clinical characteristics of the total population, *n* = 114.

	Total *n* = 114	Survivors *n* = 100 (87.7%)	Non-Survivors *n* = 14 (12.3%)	Missing Data	*p*-Value
Patient characteristics					
Age, years, mean, ± SD	87.9 ± 5.5	87.3 ± 5.4	91.5 ± 4.7	-	0.006
Gender, female, *n* (%)	82 (71.9)	72 (72)	10 (71.4)	-	1.000
Charlson Comorbidity Index score, ± SD	2.83 ± 1.7	2.73 ± 1.7	3.57 ± 1.7	-	0.119
Diabetes, *n* (%)	25 (21.9)	23 (23)	2 (14.3)	-	0.731
Chronic renal disease, *n* (%)	15 (13.2)	14 (14)	1 (7.1)	-	0.690
Chronic respiratory disease, *n* (%)	16 (14)	13 (13)	3 (21.4)	-	0.413
Chronic cardiac disease, *n* (%)	37 (32.5)	29 (29)	8 (57.1)	-	0.063
Immunosuppression, *n* (%)	17 (14.9)	15 (15)	2 (14.3)	-	1.000
ADL, mean ± SD	3.08 ± 2.1	3.25 ± 2.1	1.82 ± 1.7	-	0.020
Number of drugs, *n* ± SD	6.32 ± 3.0	6.15 ± 3.2	7.50 ± 1.8	-	0.026
Statin therapy, *n* (%)	23 (20.2)	20 (20)	3 (21.4)	-	1.000
Vaccination, *n* (%)	44 (38.6)	37 (37)	7 (50)	70 (61.4)	0.576
Influenza infection characteristics					
Influenza A, *n* (%)	107 (93.9)	93 (93)	14 (100)	-	0.594
Delay between diagnosis and the onset of symptoms, days, *n* ± SD	2.91 ± 3.5	2.94 ± 3.6	2.71 ± 2.1	-	0.408
Nosocomial infection, *n* (%)	35 (30.7)	32 (32)	3 (21.4)	-	0.545
Antiviral < 48 h, *n* (%)	48 (42.1)	45 (45)	3 (21.4)	-	0.147
Pneumonia, *n* (%)	56 (49.1)	46 (46)	10 (71.4)	-	0.092
Antibiotic prescription, *n* (%)	61 (53.5)	50 (50)	11 (78.6)	-	0.051
SOFA score, mean ± SD	1.3 (1,4)	1.12 (1.3)	2.57 (1.6)	-	0.001
Lymphopenia, *n* (%)	77 (73.3)	66 (66)	11 (78.6)	9 (7.9)	0.506

SD, Standard Deviation; ADL, Activities of Daily Living.

**Table 2 jcm-10-03521-t002:** Univariate and multivariate logistic regression analysis of factors associated with mortality.

	Univariate Analysis		Multivariate Analysis	
	OR (95% CI)	*p*-Value	OR (95% CI)	*p*-Value
Age	1.18 (1.04;1.35)	0.013	1.37 (1.05;1.79)	0.021
Female	0.97 (0.28;3.36)	0.964	3.01 (0.31;29.20)	0.342
Charlson Comorbidity Index	1.31 (0.96;1.80)	0.092	1.39 (0.78;2.48)	0.268
Diabetes	0.56 (0.12;2.68)	0,466	0.60 (0.06;5.91)	0.662
Chronic respiratory disease	1.83 (0.45;7.43)	0.401	0.27 (0.02;3.91)	0.334
Chronic cardiac disease	3.26 (1.04;10.24)	0.043	6.48 (0.56;74.69)	0.134
Immunosuppression	0.94 (0.19;4.65)	0.944	1.62 (0.16;16.40)	0.683
ADL score	0.69 (0.50;0.95)	0.027	0.36 (0.17;0.75)	0.006
Number of drugs	1.16 (0.96;1.40)	0.128	1.15 (0.83;1.61)	0.405
Nosocomial infection	0.58 (0.15;2.22)	0.426	2.17 (0.18;26.76)	0.545
Antiviral < 48 h	0.33 (0.09;1.27)	0.107	0.04 (0.002;0.78)	0.034
Antibiotic prescription	3.67 (0.97;13.94)	0.057	0.64 (0.07;6.28)	0.704
SOFA score	1.83 (1.27;2.64)	0.001	2.30 (1.07;4.94)	0.034
Lymphopenia	2.17 (0.45;10.45)	0.336	0.42 (0.04;4.03)	0.453

OR, Odds Ratios; ADL, Activities of Daily Living; SOFA, Sequential Organ Failure Assessment

## Data Availability

Data available on request due to restrictions eg privacy or ethical.

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
