# Peer review of "Risk Factors Associated with 30-Day Mortality in Older Patients with Influenza"

_jcm, 2021, doi:10.3390/jcm10163521_

Round 1

Reviewer 1 Report

For the sake of accurate disclosure, I suggest that you mention the study design (retrospective cohort design) in the abstract. In that same line of the abstract, you mention "all consecutive patients" - consecutive in what way? Maybe better to drop the word consecutive, and just say all patients with a diagnosis of influenza ....

In the introduction, first paragraph, you mention 9,000 correspondents to 11% of all-cause deaths. I guess you mean in this age group of over 75years? If so make this more clear.

Table 2. You have included the non-significant anti-retroviral <48 hrs in bold for Univariate analysis, when the p value is 0.107.

I think there is a major omission in your 'potential weakness of the study' discussion. It is highly likely that confounding by indication could be the reason for older patients not receiving antiVirals <48hrs. 

(Confounding by indication is a bias frequently encountered in observational epidemiologic studies of drug effects. Because the allocation of treatment in observational studies is not randomized and the indication for treatment may be related to the risk of future health outcomes, the resulting imbalance in the underlying risk profile between ... )

You haven't mention this as a potential weakness, that older elderly might not be prescribed antivirals, nor proven your attempts to discount this potential weakness by reporting similar proportions of prescribing antivirals in "older" and "younger" elderly patients, or by efforts to stratify to remove this potential bias.

Nor have you acknowledged that this is the only really reversible association found (although can't claim a casual association due to above and other potential biases and confounders). However, you could state that there is still benefit in quantifying non-reversible risks for the benefit of prognosis and monitoring. You have mentioned that you study is underpowered, but not really spelt out that 14 deaths (primary endpoints), even with MVA, is potential subject to lack of precision and statistical error.

Author Response

We thank the Reviewers for thoughtful reviews and the positive view of our manuscript. Please, find below our point-by-point responses to the comments and queries raised by the Reviewers. Please see corrections in green throughout the revised manuscript.

  1. For the sake of accurate disclosure, I suggest that you mention the study design (retrospective cohort design) in the abstract. In that same line of the abstract, you mention "all consecutive patients" - consecutive in what way? Maybe better to drop the word consecutive, and just say all patients with a diagnosis of influenza ....

We thank the reviewer for this comment. Please see the correction in the abstract.

  1. In the introduction, first paragraph, you mention 9,000 correspondents to 11% of all-cause deaths. I guess you mean in this age group of over 75years? If so make this more clear.

We apologize for the lack of precision. In France, the average number of observed all-cause deaths during an epidemic period is around was 85,411 and the average number of influenza-attributable deaths is estimated at 9000, i.e. 11% of the number of all-cause deaths during the epidemic period, among the French population and whatever the age. Please see the correction lines 37

  1. Table 2. You have included the non-significant anti-retroviral <48 hrs in bold for Univariate analysis, when the p value is 0.107.

The reviewer is tight. As it is mentioned in the statistical analysis section, univariable and multivariable logistic regressions were performed to examine the association between short-term mortality and others variables. Variables with a significant association in univariate analysis and/or with p-value < 0.2 were entered into a logistic regressions model for multivariate analysis. We used this threshold because it is one that we use in our team and it is a threshold that is typically used to select variables in logistic regression models.(https://www.ncbi.nlm.nih.gov/pmc/articles/PMC2633005/pdf/1751-0473-3-17.pdf, https://www.ncbi.nlm.nih.gov/pmc/articles/PMC3936971/)

I think there is a major omission in your 'potential weakness of the study' discussion. It is highly likely that confounding by indication could be the reason for older patients not receiving antiVirals <48hrs. (Confounding by indication is a bias frequently encountered in observational epidemiologic studies of drug effects. Because the allocation of treatment in observational studies is not randomized and the indication for treatment may be related to the risk of future health outcomes, the resulting imbalance in the underlying risk profile between ... ). You haven't mention this as a potential weakness, that older elderly might not be prescribed antivirals, nor proven your attempts to discount this potential weakness by reporting similar proportions of prescribing antivirals in "older" and "younger" elderly patients, or by efforts to stratify to remove this potential bias.

The reviewer is right and we thank the reviewer for this very interesting comment. A confounding bias by indication could alter the association between oseltamivir treatment and short-term mortality. In this study, we assume that the number of patient and the repartition of population, i.e. 87,9 (+/-6 5.) would not allow us to perform a stratification. However, considering the reviewer comment, we decided to correct the manuscript in the limitations section. Please see corrections in the manuscript in the discussion section (lines 311-313)

Nor have you acknowledged that this is the only really reversible association found (although can't claim a casual association due to above and other potential biases and confounders). However, you could state that there is still benefit in quantifying non-reversible risks for the benefit of prognosis and monitoring. You have mentioned that you study is underpowered, but not really spelt out that 14 deaths (primary endpoints), even with MVA, is potential subject to lack of precision and statistical error.

We thank the reviewer for this comment. Indeed, oseltamivir treatment was the only modifiable factor associated with mortality. This comment was considered and the manuscript was modified. Please see lines 251-252, line 301, and lines 313-315

Reviewer 2 Report

Influenza is a common viral disease, but in the elderly, the factors associated with short-term mortality have not been well studied. The authors investigated the factors involved in short-term mortality from influenza in the elderly and found that older age, lower ADL scores, and higher SOFA scores pose a risk of short-term mortality. They also found that oseltamivir treatment initiated within the first 48 hours was independently associated with survival.

However, in considering the aggravation of influenza and the risk of short-term death in the elderly, it can be expected that nutritional status, respiratory function and oral hygiene will be important factors in addition to the presence or absence of underlying diseases. The reviewer doubts the credibility of this data because there is not much research result on that point.

The following reviewer questions should be evaluated with additional data, or with many references to give a convincing consideration.

  1. How about is nutritional risk? For example, the status of blood albumin content seems to affect short-term mortality, but what about the truth?

  1. Is there a risk change in the subject's dietary status (i.e. an elderly can eat normally, an elderly can be eaten in liquid food or porridge, or an elderly cannot be taken by mouth, etc.) in relation to nutritional status?

  1. How about is the history of aspiration pneumonia? If elderly people have a history of aspiration pneumonia, they can easily imagine the recurrence of pneumonia and the aggravation of influenza. In that regard, were there any differences in the clinical findings of the lungs regarding the lethal and non-lethal status of influenza? Is there a difference in the number of severe cases suspected of having bacterial pneumonia?

  1. Recently, it has been reported that poor oral hygiene is associated with the severity of influenza and the risk of developing aspiration pneumonia. The oral condition of the elderly is often poor. It has also been reported that receiving professional oral care reduces the risk of developing influenza and aspiration pneumonia. From the above, it seems necessary to evaluate the oral hygiene condition.

  1. One of the causes of influenza aggravation and short-term death is pneumonia caused by secondary bacterial infection. Have you evaluated its effects?

Author Response

We thank the Reviewers for thoughtful reviews and the positive view of our manuscript. Please, find below our point-by-point responses to the comments and queries raised by the Reviewers. Please see corrections in green throughout the revised manuscript.

Influenza is a common viral disease, but in the elderly, the factors associated with short-term mortality have not been well studied. The authors investigated the factors involved in short-term mortality from influenza in the elderly and found that older age, lower ADL scores, and higher SOFA scores pose a risk of short-term mortality. They also found that oseltamivir treatment initiated within the first 48 hours was independently associated with survival. However, in considering the aggravation of influenza and the risk of short-term death in the elderly, it can be expected that nutritional status, respiratory function and oral hygiene will be important factors in addition to the presence or absence of underlying diseases. The reviewer doubts the credibility of this data because there is not much research result on that point. The following reviewer questions should be evaluated with additional data, or with many references to give a convincing consideration.

  1. How about is nutritional risk? For example, the status of blood albumin content seems to affect short-term mortality, but what about the truth?

The reviewer is correct. Nutritional status is associated with mortality (Please see reference 16 in the manuscript and Katona and Katona-Apte Clinical Infectious disease 2008). However, it is unclear whether malnutrition plays a causative role in adverse outcomes or if it serves as a marker for decreased health status, chronic inflammation, hormonal and metabolic derangements, more advanced comorbidities, or frailty (Abugroun American Journal of Medicine 2020). In this study, the retrospective design was associated with missing data (previous weights, sizes, and serum albumin) and thus, unfortunately, we could not consider it, as it was mentioned in the discussion section (line 285). Moreover, it should be noted that serum albumin is not a good marker of malnutrition, especially during an hypercatabolic episode (as infectious disease), and that’s why serum albumin was recently removed from the diagnosis criteria of malnutrition (Cederholm Clin Nutr 2019). However, this comment is crucial and we decided to highlighted even more these points in the discussion section and the discussion (lines 288-290) and in the conclusion (line 321).

  1. Is there a risk change in the subject's dietary status (i.e. an elderly can eat normally, an elderly can be eaten in liquid food or porridge, or an elderly cannot be taken by mouth, etc.) in relation to nutritional status?

The reviewer is correct, during hospitalization, there may be dietary change that could be associated we malnutrition. In a geriatric ward, dietary change are closely and daily monitored. These adaptations have proven to be beneficial for patients and were carried out in the same way for all patients, i.e the survivors and the non survivors. In this study, for the non survivors and the survivors, the malnutrition risk was considered and treated as it is recommended. Once again, denutrition status and dietaty status are crucial points and as it was detailed in the previous answer, we proposed a correction in the manuscript (lines 288-290)

3.How about is the history of aspiration pneumonia? If elderly people have a history of aspiration pneumonia, they can easily imagine the recurrence of pneumonia and the aggravation of influenza. In that regard, were there any differences in the clinical findings of the lungs regarding the lethal and non-lethal status of influenza? Is there a difference in the number of severe cases suspected of having bacterial pneumonia?

The reviewer is correct, aspiration pneumonia could aggravate influenza. That’s why we decided to considered RT-PCR positive episode, in order to be sure that patients had influenza associated pneumonia. Moreover, because of the retrospective design we could not monitor and collected inhalation and/or aspiration episodes with or without pneumonia.

The reviewer is right, bacterial co-infection are frequent and could increase mortality. An autopsy series from the 2009 H1N1 influenza pandemic found evidence of bacterial co-infection in about 30% of deaths (Shieh Am J Pathol 2010). That’s why, it is recommended to prescribe antibacterial treatment for adults with clinical and radiographic evidence of pneumonia and with positive test for influenza (Metlay American Journal of Respiratory and Critical Care Medicine). In this study, we observed a very high prevalence of pneumonia and a very high correlation between antibiotic prescription and pneumonia diagnosis (pearson correlation coef >0,7). In detail, pneumonia diagnosis and antibiotic prescription were observed, respectively, in 50,0% and 46,0% of patient in the survivor group, and in 71,4% and 78,6% of patients in the non-survivor group. In our study, none patient had positive bacteriological identification. This is not surprising because, in our study, few patients had severe form (mean SOFA score of 1.3) and sputum culture and/or bronchoalveolar sample collection are not recommended in non-severe patient (Metlay American Journal of Respiratory and Critical Care Medicine). As a reminder, sepsis (bacterial or viral sepsis) should be considered as life-threatening organ dysfunction caused by a dysregulated host response to infection. In clinical practice, organ dysfunction can be represented by an increase in the Sequential [Sepsis-related] Organ Failure Assessment (SOFA), whatever the cause (Singer JAMA 2016). Indeed, the Sequential [Sepsis-related] Organ Failure Assessment (SOFA) score is a consensual score associated with sepsis severity and a score of 2 points or more, is associated with an in-hospital mortality greater than 10% (Singer JAMA 2016).

Finally, we completely agree with the reviewer comment concerning the increased mortality associated with suspected bacterial co-infection and the severity of sepsis. That’s why we decided to included antibiotic prescription and the SOFA score in the multivariable model (Table 2).

  1. Recently, it has been reported that poor oral hygiene is associated with the severity of influenza and the risk of developing aspiration pneumonia. The oral condition of the elderly is often poor. It has also been reported that receiving professional oral care reduces the risk of developing influenza and aspiration pneumonia. From the above, it seems necessary to evaluate the oral hygiene condition.

The reviewer is right, good oral hygiene has been recognized as a means to prevent airway infections in patients, especially in those over the age of 70 (Sjögren J Am Geriatr Soc 2008) and, since many years, is it well known that oral hygiene is associated with the risk of developing pneumonia (Muder Am J Med 1998, Mylotte Am Fam Physician 2009, Ewig S et al. Thorax 2012). Two recent Cochrane reviews (Liu Cochrane database Syst Rev 2018, Hua Cochrane database Syst Rev 2016) highlighted this, even if the authors found no high-quality evidence to determine which oral care measures are most effective for reducing pneumonia.

Recently, the COVID-19 pandemia highlights the fact that oral hygiene could be associated with pneumonia mortality but with a relative low level of evidence (Imai Int J Mol Sci 2021). However, few studies have explored the association between oral hygiene and mortality, mainly focused on non-communicable disease (heart disease, diabetes, kidney disease) and long-term mortality (Mitzutani Nature 2020, Genco Nat Rev Cardiol 2010, Graziani J Clin Periodontol 2018). Finally, to our knowledge, there is no consensus and a low level of evidence that oral hygiene could be associated with influenza short-term mortality, although there is a fundamental rational (Kamio Cell Mol Life Sci 2015). Indeed, a recent systematic review (Data Dent J 2021), focused on the impact of oral health on influenza virus infection, highlighted the necessity to investigate these question.

  1. One of the causes of influenza aggravation and short-term death is pneumonia caused by secondary bacterial infection. Have you evaluated its effects?

The reviewer is right, many factors are associated with influenza mortality and could alter the short-term mortality, as it is mentioned in the introduction section [presence of underlying diseases (13), including diabetes (5,8) immunosuppression (4,7,8,17), chronic respiratory disease (6,7), chronic cardiac disease (6,14), renal disease (6), cancer (7,12) ; number of drugs (6,16) ; malnutrition (16) ; recent infectious disease (16) ; severity of illness (4,5,10,12,14,17) including bacterial co-infection (9) and noso-comial infection (9) ; delay in receiving adequate treatment and admission in an inten-sive care unit (7,8,11), delay in the initiation of antiviral treatment (within the first 48 hours) (8,9,13,17) ; number of consultations in the past year (15)”] and in the discussion section [“we did not observe an association between the short-term mortality and the fol-lowing factors that was previously associated with short-term mortality in patients with influenza: presence of underlying diseases (13), including diabetes (5,8), immu-nosuppression (4,7,8,17), chronic respiratory disease (6,7), chronic cardiac disease (6,14), renal disease (6), cancer (7,12) ; number of drugs (6,16) ; malnutrition (16) ; re-cent infectious disease (16) ; including bacterial co-infection (9) ; nosocomial infection (9) ; number of consultations in the past year (15)”]. The reviewer is correct, secondary aggravation and or death could be caused by secondary bacterial infection but also to an aggravation of comorbid disease. That’s why we decided to consider short-term all-cause mortality in this study. We thank the reviewer for this comment and we apologize for the lack of precision in the manuscript. Thanks to this comment, we decided to provide more detail in the manuscript in the materials and methods section and in the discussion. Please see correction in the manuscript on lines 72, 303, 304 and 306.

Round 2

Reviewer 2 Report

The authors accurately answered the reviewers' questions and revised the manuscript to reflect them. The reviewer is satisfied with the content of the revised manuscript and are willing to accept it as the article.